# Meta-Analysis Reveals Challenges and Gaps for Genome-to-Phenome Research Underpinning Plant Drought Response

**DOI:** 10.3390/ijms232012297

**Published:** 2022-10-14

**Authors:** Anthony E. Melton, Stephanie J. Galla, Carlos Dave C. Dumaguit, John M. A. Wojahn, Stephen Novak, Marcelo Serpe, Peggy Martinez, Sven Buerki

**Affiliations:** Department of Biological Sciences, Boise State University, Boise, ID 83725, USA

**Keywords:** arid habitats, climate change, drought stress, green plants, genome-to-phenome, text mining

## Abstract

Severe drought conditions and extreme weather events are increasing worldwide with climate change, threatening the persistence of native plant communities and ecosystems. Many studies have investigated the genomic basis of plant responses to drought. However, the extent of this research throughout the plant kingdom is unclear, particularly among species critical for the sustainability of natural ecosystems. This study aimed to broaden our understanding of genome-to-phenome (G2P) connections in drought-stressed plants and identify focal taxa for future research. Bioinformatics pipelines were developed to mine and link information from databases and abstracts from 7730 publications. This approach identified 1634 genes involved in drought responses among 497 plant taxa. Most (83.30%) of these species have been classified for human use, and most G2P interactions have been described within model organisms or crop species. Our analysis identifies several gaps in G2P research literature and database connectivity, with 21% of abstracts being linked to gene and taxonomy data in NCBI. Abstract text mining was more successful at identifying potential G2P pathways, with 34% of abstracts containing gene, taxa, and phenotype information. Expanding G2P studies to include non-model plants, especially those that are adapted to drought stress, will help advance our understanding of drought responsive G2P pathways.

## 1. Introduction

Anthropogenically driven climate change is expected to exacerbate drought conditions in landscapes around the world, currently affecting up to 85% of the world’s population [1,2,3], reviewed in [4]. Climate change is causing an increase in mean annual temperatures, which could potentially increase by 2 °C globally by 2050, and nearly 6 °C by 2100 [5]. Changes to precipitation patterns are also expected, with some regions predicted to experience severe flooding or drought with greater frequency [6]. For example, a recent study by Williams et al. [6] revealed that southwestern North America has recently experienced the largest megadrought in the past 1500 years, which is largely attributable to anthropogenic climate change magnifying natural climatic variability [7]. The expected increases in drought due to higher temperatures and reduced precipitation are significant threats to agriculture, and natural communities and ecosystems [8]. The capacity of crop plants to adapt to new drought conditions is essential for providing food security as human population growth continues [9,10] (more text on this topic and literature review is provided in the Appendix A given in Appendix A).

Beyond agricultural plant species, there is great interest in understanding how natural plant species and communities respond to intensifying drought stressors to preserve stable and functioning ecosystems in a changing world reviewed in [11]. For many plant species, the ability to naturally disperse towards areas with optimal niche space is limited by factors such as short seed-dispersal distances and habitat loss and fragmentation. As a result of these or other limitations, geographic shifts may not keep pace with new and altered habitats produced by climate change [12,13]; reviewed in [11]. This is especially true when other anthropogenic threats (e.g., habitat loss, fragmentation, and reductions in population size) occur jointly with climate change [14,15]. Therefore, identifying the mechanisms allowing plants to persist as climate change, and especially drought, proceeds and intensifies is paramount for evaluating adaptive capacity of these populations and designing and implementing best management practices for a sustainable future [16].

There is an urgent need to identify mechanisms of response to the stresses caused by climate change, particularly drought. Reductions in the cost of high-throughput nucleotide sequencing over the past two decades [17] have made it possible to investigate the genomic underpinnings of these responses in non-model species at a much greater scale and rate. While the genomic basis of drought adaptation has been explored in well-resourced crop plants (e.g., corn, *Zea mays*; rice, *Oryza* spp.; wheat, *Triticum* spp.) [18,19,20], plant model systems (e.g., *Arabidopsis thaliana*) [21,22], and forestry species reviewed in [23], fewer non-model or natural plant systems have been explored to the same extent. This disparity is reflected in the number of genomic resources that are readily available for plant species. For example, of the 243 unique and chromosome-level plant reference genomes available on NCBI (as of December 2020), only 20% are from species of non-human use, as classified by the Kew Gardens [24]. We anticipate this disparity in genomic resources may be due to several barriers to conducting conservation genomic research (i.e., the ‘conservation genomics gap’) [25,26], including the cost of sequencing and assembling the large and complex genomes that many plants possess [27,28]. Therefore, a cost-effective and targeted approach to finding genes associated with drought response is required to fill this gap for natural populations.

Reviewing the state of research focused on G2P pathways will allow for focused hypothesis-testing research on non-model plant species for which little genomic data is available. Mining genomic and phenomic data across a wide phylogenetic breadth will allow for a better understanding of the diverse and complex plant responses to drought and provide novel opportunities for crop improvement. While mining methods applied to publicly available datasets and publications can advance our understanding of G2P processes, there is a stark lack of infrastructure available to perform such searches and analyses in a thorough fashion. While many genetics and genomics journals now require that sequence data be publicly available in a repository (e.g., PLoS ONE enacted a “Data Availability Statement” requirement in March of 2014 [29]), this has not always been the case and many studies lack connectivity to an available dataset [30]. This creates an urgent need to increase database connectivity to aid in increasing reproducibility and allowing for meta-analyses.

To make efficient and rapid gains in understanding the genomic-basis of plant response towards drought, we developed a literature and database search pipeline and performed a meta-analysis to identify focal taxa of research on genes underpinning drought-stress response across the primary literature. Abstracts from these articles were mined using the R package ‘G2PMineR’ [31] to produce networks connecting genes to phenotypic traits associated with drought-stress responses. Identifying key gene networks and drought-adapted species for which genomic resources may be available for will provide potential routes forward in drought G2P research. Databases containing genetic and taxonomic data related to these publications were queried to identify patterns of genome-to-phenome research and to assess connectivity of these publications and databases. This methodology enables investigations into how genes are associated with physiological processes and drought-stress response strategies across a broad range of taxa. This study gives a broader understanding of drought G2P research in plants. It has generated a list of genes and associated phenotypes to provide a foundation for studying the genomic and phenotypic basis of drought tolerance of less-studied natural plant species worldwide.

## 2. Results

### 2.1. Literature Mining

The PubMed search query used in this study resulted in the acquisition of 7730 abstracts from papers published from January 1998–December 2020. Abstract mining using ‘G2PMineR’ [31] produced a total of 2642 consensus abstracts having data for taxa, genes, and phenotypes (Appendix A).

### 2.2. Taxa

Within the ‘G2PMineR’ consensus abstracts, 497 species or subspecies, representing 296 genera and 98 families (91 angiosperm families, four gymnosperm families, two fern families, and one lycophyte family; Appendix A) were identified. The top reported species is *Arabidopsis thaliana* L., with 641 mentions in the consensus abstracts (Figure 1A). Other highly studied species include *Oryza sativa* L. (Poaceae; 259 mentions), *Triticum aestivum* L. (Poaceae; 131 mentions), *Zea mays* L. (Poaceae; 87 mentions), and *Glycine max* (L.) Merr. (Fabaceae; 77 consensus mentions). The most studied genes are reported in a wide breadth of taxa, indicating that they play a role in drought response across diverse plant lineages (Figure 1D). Poaceae had the highest representation, with 73 taxa mentioned in consensus abstracts. Other highly represented families included Fabaceae (66 taxa) and Solanaceae (29 taxa). Poaceae species were the most mentioned in consensus abstracts (828 mentions), followed by Brassicaeae (800 mentions) and Fabaceae (417 mentions). Within the consensus abstracts, 414 (83.30% of species mentioned) are classified for human use (Appendix A). Of the plants with human uses according to the Kew Gardens Checklist of Useful Plants [24], the most common uses were medicinal (318 taxa), environmental (251 taxa), food (229 taxa), materials (218 taxa), and gene sources (208 taxa), with these categories not being mutually exclusive. Of the plants identified in consensus abstracts, 103 (20.7% of consensus abstract taxa) were listed as having chromosome level assemblies per NCBI database (Appendix A).

Locality data were successfully downloaded for 106 species mentioned in the total abstracts lacking human use according to the Kew Gardens Checklist of Useful Plants [24]. Of these, 31% (33 of 106 species) were found to occur in hyper-arid environments based on aridity index values (AI < 0.05), though none of these species were included in the consensus abstracts (Appendix A). One occurrence for *Tillandsia ionantha* Planchon was found in northern Africa and was excluded as it is native to Central America and Mexico and the accuracy of this occurrence could not be confirmed [32]. The most mentioned plant species lacking from hyper-arid environments lacking human use per Kew list were: *Cleistogenes songorica* (Roshev.) Packer (Poaceae; eight abstracts), *Caragana korshinskii* Kom. (Fabaceae; seven abstracts), *Malus prunifolia* (Willd.) Borkh. (Rosaceae; seven abstracts), *Aeluropus littoralis* (Gouan) Parl. (Poaceae; six abstracts), *Malus sieversii* (Ledeb.) M.Roem. (Rosaceae; six abstracts), *Boea hygrometrica* (Bunge) R. Brown (Gesneriaceae; five abstracts), *Reaumuria soongorica* (Pall.) Maxim. (Tamaricaceae; five abstracts), *Alhagi sparsifolia* Shap. (Fabaceae; four abstracts), *Rosa hybrida* (Rosaceae; four abstracts), and *Chorispora bungeana* Fisch. & C.A. Mey (Brassicaceae; three abstracts).

### 2.3. Genes

A total of 1634 genes, including single-copy genes and gene groups (e.g., *WRKY*) associated with drought response phenotypes were found in the ‘G2PMineR’ consensus abstracts (Appendix A). The top-10 cited genes included *oasC* (O-acetylserin (thiol) lyase isoform C; 345 mentions), *cysC* (cysteine synthase C1; 203 mentions), *NSY* (chloroplastic neoxanthin synthase; 183 mentions), *CITRX* (Cf-9-interacting thioredoxin; 181 mentions), *NAC* (NAM, ATAF, and CUC transcription factors;128 mentions), *WRKY* (WRKY-domain containing transcription factors; 127 mentions), *SODCC.1* (superoxide dismutase [Cu-Zn] 1; 118 mentions), *MYB* (MYB DNA-binding domain-containing transcription factors; 105 mentions), *PLA3* (probable glutamate carboxypeptidase; 97 mentions), and *LEA* (late embryogenesis abundant proteins; 87 mentions; Figure 1B).

### 2.4. Phenotypes

A total of 118 phenotypes from our curated drought phenotype lexicon were identified in the consensus abstracts (Appendix A). The top 10 phenotype words within the consensus abstracts were “drought” (1307 mentions), “growth” (940 mentions), “development” (845 mentions), “salt” (823), “abscisic acid” (726 mentions), “cold” (551 mentions), “salinity” (439 mentions), “root” (430), “seed” (357 mentions), and “stomata” (329 mentions) (Figure 1C). The top drought strategy for which phenotypes were mentioned was the “avoidance” strategy, with 6383 mentions. For other drought strategies, the total mentions for phenotypes of a given group was: 1751 for “detection”, 2945 for “recovery”, 3570 for “general stress”, 3703 for “escape”, and 5765 for “tolerance”.

### 2.5. Gene to Phenotype Interactions

Of the top gene and phenotypes, *oasC* and general phenotype “drought” had the greatest connectivity (Figure 2). *oasC* also had high connectivity with “abscisic acid”, “development”, and “growth”. *CITRX*, *WRKY*, *NAC*, *NSY*, and *cysC* genes were also highly connected to the phenotype “drought”. Most genes were associated with multiple drought-response strategies: a total of 759 genes (~48% of genes identified in consensus abstracts) were associated with phenotypes of all major drought-response strategies, 366 (21%) were associated with “avoidance”, “escape”, “recovery”, and “tolerance”, and 120 (8%) of consensus genes were associated with “avoidance”, “detection”, and “tolerance” (Figure 3). These genes were associated with a variety of gene ontology (GO) categories, though the most common GO categories were abscisic acid response (GO:0009737, 73 genes), water deprivation response (GO:0009414, 66 genes), and abscisic acid activated signalling pathways (GO:0009738, 62 genes).

### 2.6. Assessing Connectivity of Sequence Data and Taxonomy Databases

In total, 1712 (22.15%) of the 7730 abstracts were associated with sequence data and 1865 (24.13%) were associated with taxonomic data on NCBI. Only 1624 (21.01%) abstracts had connectivity to GenBank databases, for both taxonomy and gene data, for each publication, per assessment using ‘rentrez’ [33].

A total of 583 taxa were found to be associated with the PubMed IDs using ‘rentrez’. Of these, five taxa were at the familial level, 25 taxa were at the generic level, and 15 taxa were identified as non-plants. The publications included one to 152 species/taxa, with all but six focusing on less than 10 taxa. A total of 192 taxa were shared with the taxa identified in the abstract mining search.

A total of 423 genes with the GenBank database were associated with the PubMed IDs. Few studies were associated with many nucleotide sequences (e.g., 37 studies were associated with greater than 1000 sequence accessions). The most common genes from the studies with sequence accessions in GenBank included *oasC* (274 accessions), *NAC* (37 accessions), and *WRKY* (35 accessions). A total of 234 genes identified from sequences were shared with those identified in the abstract mining search.

Due to a lack of phenotypic data in NCBI accessions, phenotype data had to be mined from abstracts. This produced a total of 114 phenotypes identified from abstracts that had associated sequence and taxonomy data, all of which were included in the abstract mining search.

## 3. Discussion

### 3.1. Trends and Gaps in Plant Drought Research

Only 21% of studies (1624 of 7730 abstracts) had connectivity to NCBI databases for gene sequences or taxa. Of these, 1533 studies focused on one taxon, with the greatest number of taxa studied in any one study being 152. Most studies also focused on only one sequence, while some were associated with over 100,000 sequences. Presumably, the studies with greater numbers of sequences but only one taxon associated with them in databases represent expression characterization studies with many replicates of the taxon (e.g., [34,35]). The most studied species from papers with both sequence and taxon database connectivity was *Arabidopsis thaliana*, with 730 studies on this species being linked to both sequence and taxon data. While these sequence data were associated with plant-oriented studies, several taxa associated with these data were not plants. For example, a study on calcineurin B-like proteins in *Arabidopsis* is also associated with sequence data from *Rattus norvegicus* (brown rat), which was used to test for interactions between calcineurin B-like proteins [36]. Some bacteria species were identified, which were mostly used in gene transformation or cloning experiments (e.g., [37]). While the ‘G2PMineR’ workflow is naïve to whether a given gene mentioned in abstracts is characterized from an expression study or from genomic sequence, this does not affect the interpretation of the workflow results. The ‘G2PMineR’ abstract mining workflow was more successful at making G2P connections, with 34% of abstracts containing gene, taxa, and phenotype data. Consensus abstracts contained all requisite information for making associations between taxa, genes, and phenotypes, which is apparently not possible to any meaningful level using database associations. Characterization of genes, species, and phenotypes involved in drought tolerance was more successful using the G2PMineR pipeline, compared to simply using manual searches. For example, 1634 genes were identified using ‘G2PMineR’ while only 423 genes associated with the PubMed abstract set were found within GenBank. Interestingly, only 234 genes were shared within the two datasets (Figure 4).

The ‘G2PMineR’ workflow illustrates the current state of drought response research among plant species regarding different plant species under investigation. Approximately 83% of plant species identified in plant drought abstracts represent either model species, crop species, or species of human use. The top species identified was the model plant species *Arabidopsis thaliana* L., which is not surprising because *Arabidopsis* is the primary model plant for genetic research and has a well-assembled and annotated genome with vast genetic resources available for G2P research [38]. In addition to *Arabidopsis thaliana*, crop species such as *Oryza sativa* L. (rice) or *Zea mays* L. (corn) have been the primary focus of drought research, which is expected given that climate change, especially droughts, poses a threat to agriculture [39]. A large phylogenetic breadth of species was found in our search across the plant kingdom. The taxa identified in the consensus abstracts represented 98 green plant families, with 95 of the represented plant families being seed-plant families. While species of many families were included in the consensus abstracts, these species were generally biased towards families with model plant species and economically important species, such as Poaceae, Fabaceae, and Solanaceae (Figure 3). Outside of seed plants, only two fern (Hymenophyllaceae and Pteridaceae) and one lycophyte (Selaginellaceae) families were represented. Because of this bias, researchers may have only a narrow understanding of how plants adapt and respond to drought stress. The use of genetic model plants and crop plants in any genomic or adaptive capacity research is of great importance and may be broadly applicable across a wide taxonomic breadth (e.g., [40]). However, non-model plants, plants of arid ecosystems, and plants native to threatened ecosystems need broader inclusion in G2P drought research. Studying the G2P drought response pathways of plants that are well-adapted to dry environments, such as the taxa highlighted in Appendix A, will contribute to our understanding of the various metabolic and physiological pathways that increase drought tolerance, and could potentially be used for crop improvement. While a similar number of taxa were identified using ‘rentrez’ and ‘G2PMineR’ (391 versus 305, respectively), there was little overlap, with only 192 (~28% of all taxa identified from both abstract and database mining) being found in both abstract sets.

Relatively few plant species found in consensus abstracts (103 of 497; 20.7% of consensus abstract taxa) had high quality genome assemblies (i.e., assembled to chromosome level) available, with a bias towards model organisms. With decreasing costs, increasing quality, and the ease of generating whole genome assemblies for non-model organisms [17], there are more opportunities to include non-model and natural plant species in G2P research. We anticipate that the efforts of individual research labs and genomic consortiums (e.g., 10,000 Plant Genomes Project [41]) will help bridge this gap in the coming years.

Few of the species targeted in G2P research thus far occur in arid or drought-prone areas. Only 16.70% of species and subspecies identified in the consensus abstracts lacked human use per the Kew Gardens checklist. Of the non-model or non-frequently used plant species identified across all abstracts, roughly 33% occur in arid regions (Appendix A). While the study of model and crop plant species is important, as these species have the financial resources and infrastructure available for in-depth genomic research, more effort needs to be directed to study species in ecosystems in which drought-adapted species have evolved. Among these, the species identified from all abstracts native to hyper-arid habitats and well-adapted to severe droughts belong primarily to families for which genomic resources are readily available (e.g., Brassicaceae, Fabaceae, Poaceae, and Rosaceae). Consequently, these species are good candidates for G2P research, potentially revealing new mechanisms to improve drought tolerance in con-familial crop species. Given that no species in the consensus abstracts occur in areas where plants must be highly adapted to drought-stress, this is an area that needs more attention by researchers.

### 3.2. Drought Response Strategies

In the consensus abstracts, the most-studied responses fell under the drought avoidance strategy, which includes the ability of plants to reduce drought stress by minimizing water loss and increasing water uptake [42]. This observation is consistent with the high occurrence of keywords such as abscisic acid (ABA) and osmotic adjustment. ABA is well known to induce stomatal closure and increase the root to shoot ratio, while osmotic adjustment allows plants to extract more water from drying soil profiles [43,44,45]. Many of the top genes mentioned in consensus abstracts comprise transcription factors (e.g., *NAC*, *MYB*, *WRKY*) that affect stomatal activity [46,47], which agrees with the high occurrence of avoidance strategy words in consensus abstracts. In addition to avoidance, strategies that reduce drought-induced cellular damage due to accumulation of toxic molecules, particularly “synthesis of osmoprotectants” and “antioxidant production”, were also often mentioned in consensus abstracts. Relatively fewer studies were associated with mechanisms of drought detection or escape and recovery from drought, suggesting a need for more research in these areas. While drought was the main focus of this G2P investigation, other stressors like cold and salt were commonly mentioned. This result reflects the fact that plants exhibit specific, as well as shared, responses to different abiotic stresses [44]. Some of the damage caused by various abiotic stresses (e.g., increased ROS production and dehydration) can be similar [48,49]. With shared responses, plants can use many of the same genes to minimize damage and cope with multiple stresses. We anticipate that as plant G2P research continues, findings across stress types (e.g., temperature, light, salt, heavy metals) will be applicable to one another and overall enhance our understanding of stress response.

### 3.3. Candidate Genes for Future G2P Research

Highly reported genes signal broad function in stress response pathways, particularly those in osmoprotectant and antioxidant pathways, and were associated with taxa of many of the plant families represented in this study. Of the top ten most referenced genes, *oasC*, *cysC*, *LEA*, *NAC*, and *SODCC.1* have functions associated with, but not limited to, osmoprotectant and oxidative pathways [50,51,52]. Other commonly referenced genes function in carotenoid metabolic pathways involved in synthesizing ABA precursors or antioxidants such as β-Carotene [53,54]. Many of the genes in the consensus abstracts include transcription factors, such as *WRKY, NAC*, and *MYB*, which have widespread functions in stress response pathways, including stomatal closures [46,47]. These transcription factors act like ‘switches’ to regulate expression of many genes and are predicted to be one of the most important mechanisms for plant drought response and increase our capacity to engineer more resilient plant species [55,56].

Most of the top occurring genes were found to be highly connected to the phenotype “drought” in the consensus abstract, which is to be expected given our initial search query. However, our top reporting genes (e.g *SODCC.1*., *oasC*) showed high connectivity with many phenotypes of different stresses (e.g., “salt”, “cold”), response strategies (e.g., “growth”, “development”), and phytochemical pathways (e.g., “abscisic acid”, “proline”, “salicylic acid”). This indicates how genes with more targeted functions (e.g., osmoprotectants and antioxidants) can play general roles in stress response. Indeed, DroughtDB by Alter et al. (2015) [57] shows how some genes can play multiple roles across drought response. As G2P plant drought research continues to expand, we expect the functions of genes will become better characterized. In the meantime, this meta-analysis provides a list of candidate genes identified across taxa that can be broadly explored in species of interest using various methods, such as the mining of draft genomes (e.g., [58]).

While there is a large focus on phenotypes associated with the drought-avoidance and -tolerance strategies in the consensus abstracts, genes contributing to phenotypes associated with all of the drought response strategies were highly referenced in the consensus abstracts (Figure 3). These include all the most studied genes identified in this meta-analysis. This indicates that while there are some gaps in focal phenotypes and drought response strategies, many genes underpinning these phenotypes and response strategies have likely been well characterized and could readily be transferred to and used in studies of many other drought response strategies if they are not already in use.

### 3.4. Limitations of Literature and Database Mining

While we have been able to identify numerous genes and species used in the study of drought stress response, there are limitations to the text mining methods employed here. One primary limitation is that the text mining methods outlined here mine only published abstracts. Given publisher constraints, such as word count, abstracts often give broad summaries and may not list specific genes or species which were studied. For example, studies on expression patterns and functional pathways for gene families are quite common and these publications may only provide the gene family or group being studied (e.g., *WRKY*) and not specific members of these gene families. While measures were taken to account for such studies (e.g., including the gene family/group name in the searches, as well as limiting the final consensus abstract genes to and grouping by SwissProt families), conclusions cannot be drawn on which specific genes within the gene families confer improved drought-stress responses without a more in-depth reading of Results and Discussions sections of published literature.

Furthermore, there is a disconnect between publications and the databases in which sequence data are stored: only 21% of abstracts were associated with sequence and taxonomy data in NCBI databases. Some of this may be due to the timing of publication (e.g., older publications are less likely to have sequence data deposited into GenBank (Appendix A)), but as noted by searching GenBank, even recent publications are not always linked to the genes, genomes, or proteins within databases. G2P research and meta-analyses/reviews such as this one will benefit from greater connectivity between sequence databases and publications (e.g., linking a sequence in GenBank to a publication by including an identification number such as a PubMed ID).

## 4. Conclusions

Gaps have been identified in the plant taxa used for G2P drought research. Most effort in this area has focused on model plants and not on plants naturally occurring in drought-prone ecosystems. Several species of hyper-arid ecosystems, with close relatives whose genomes have been sequenced, were identified providing species for future research on drought response G2P pathways. Many commonly studied genes contribute to multiple drought response strategies, even though the research focused on a subset of these strategies. These genes provide excellent starting points for G2P research on non-model plants with few genomic resources. Given the gaps in focal taxa and the availability of genomic and genetic resources, especially for commonly studied genes and gene families, this presents a clear path for rapidly advancing drought response research in the future.

## 5. Materials and Methods

### 5.1. PubMed Search

The PubMed (https://pubmed.ncbi.nlm.nih.gov/ (accessed on 3 January 2021)) database was chosen for our literature search, as it includes all genetic and genomic studies hosted through NCBI’s GenBank and is open source. We developed search query strings in PubMed to find all publications relevant to literature pertaining to genes underpinning plant drought. We manually evaluated results of several query strings for content and verified targeted searches using the *AbstractClusterMakeR* and *MembershipInvestigateR* functions in the R package ‘G2PMineR’ (https://buerkilabteam.github.io/G2PMineR_Web/ [31]). The results of the final query, completed in December 2020, included publications between the years 1998 and 2020 with the terms “plant AND drought AND (resistance OR tolerance OR response OR avoidance OR resilience OR susceptibility OR escape) AND gene”. Figure 5 shows the workflow for analyses performed on the downloaded abstract data.

### 5.2. Literature Mining and Analyses

All abstracts were mined for species studied, genes referenced, and phenotypes expressed using the R package ‘G2PMineR’ [31]. We used the function *SpeciesLookeR* to mine abstracts for all species characterized as ‘green plants’ (Class Viridiplantae) within the Global Biodiversity Information Facility (GBIF; GBIF.org) database. To ensure that our assessment included up-to-date taxonomies, we used the ‘taxize’ R package [59] to identify synonyms and the primary accepted scientific name for each taxon used in the abstracts. We further determined how many plants are assigned as being useful to humans (i.e., plants for foods, poisons, medicines, environmental uses, social uses, fuels, or materials) per a list generated by the Kew Gardens [24], to assess bias towards non-model organisms. NCBI (https://www.ncbi.nlm.nih.gov/genome/ (accessed on 5 January 2021)) was queried for species with chromosome level genome assemblies to determine how many of the species mentioned in abstracts included here have high-level genome assemblies. To determine how species of interest identified from these abstracts are geographically distributed, GBIF (see [60] for the GBIF dataset) and iDigBio (https://www.idigbio.org/ (accessed on 7 April 2021)) were queried for all species lacking human use per the Kew Gardens Checklist of Useful Plants [24]. This search was restricted to these non-model and/or crop plants, as species of agriculture and horticulture are often logged in public databases through citizen scientists, due to biases in observations being made in or near areas of human activity, and many of these occurrences do not represent natural occurrence [61]. Occurrence data were cleaned to remove data lacking complete coordinates, having unlikely coordinates (e.g., 0,0), and lacking environmental data using the ‘scrubr’ R package [62] using aridity index (AI) as a reference (https://cgiarcsi.community/data/global-aridity-and-pet-database/; (accessed on 14 February 2021) [63]). Occurrence data were used to extract AI values, which were then used to identify species that are adapted to low-water stress, particularly those found in hyper-arid (AI < 0.05) environments (all occurrence data used to identify species of hyper-arid environments are available at DOI:10.5281/zenodo.7153278).

To understand which genes are being described in each study, the ‘G2PMineR’ function *GenesLookeR* was used to search abstracts for 40,435 genes, their putative gene families, and ontologies as listed in SwissProt for Viridiplantae, as of July 2020. To capture genes and broad family groups (e.g., *WRKY* as opposed to *WRKY1*) that were cited in abstracts, but not always captured in SwissProt, we generated an additional lexicon of genes and gene family names from the DroughtDB [57], the Plant Transcription Factor Database (http://planttfdb.gao-lab.org/ (accessed on 31 August 2020)), and those reported in previously published plant drought gene reviews (e.g., [9,64,65]). The ‘G2PMineR’ function *SynonymReplaceR* is used to replace gene synonyms with accepted gene names from SwissProt. See Appendix A for a complete list of gene and gene family names identified in extracted abstracts.

Finally, we used the ‘G2PMineR’ function *PhenotypeLookeR* to search abstracts for phenotypic words of interest, as defined by a curated glossary from the Missouri Botanical Garden [66]. To make a lexicon that is specific to drought, we curated a drought phenotypic lexicon using terminology from Farooq et al. (2012) [67], De Micco & Aronne (2012) [68], and Vilagrosa et al. (2012) [69]. To test whether genes were associated with phenotypes that represented different drought response categories (Appendix A), all phenotypes were manually categorized as being relevant to drought detection, resistance (i.e., escape, avoidance, and tolerance), and/or recovery. See Appendix A for a full list of phenotypes and categories used.

All downstream analyses for species, genes, and phenotypes were conducted for abstracts that had all three search items represented using the *ConsensusInferreR* function within ‘G2PmineR’. In order to elucidate the link between genotypes and phenotypes, networks were developed using the ‘G2PMineR’ function *PairwiseDistanceInferreR*, which generates a pairwise distance matrix between each gene and phenotype using the number of shared abstract matches. To place taxa represented in the consensus abstracts into a phylogenetic context, families of all represented taxa were placed into a family level flowering plant phylogeny [70]. Family placement for each taxon was acquired from NCBI using the ‘taxize’ R package [59]. Consensus abstract mentions, taxa, genes, and phenotypes represented were then tabulated for each family.

### 5.3. Assessing Connectivity to Databases

The R package ‘rentrez’ [33] was used to mine gene and taxonomy information for publications that have both GenBank sequence and taxonomy data linked to the respective PubMed ID. To ensure that taxonomy was standardized across comparisons, the R package ‘taxize’ [59] was used to evaluate taxa and replace any synonyms with the accepted taxon name. Due to a lack of standardized formatting of data within fasta headers, ‘G2PMineR’ [31] was used to mine gene names from headers using ‘G2PMineR’ databases. Phenotype data for abstracts with both gene and taxonomy data database associations were then subset from the consensus abstract phenotype data for comparisons to abstract mining results.

## Figures and Tables

**Figure 1 ijms-23-12297-f001:**
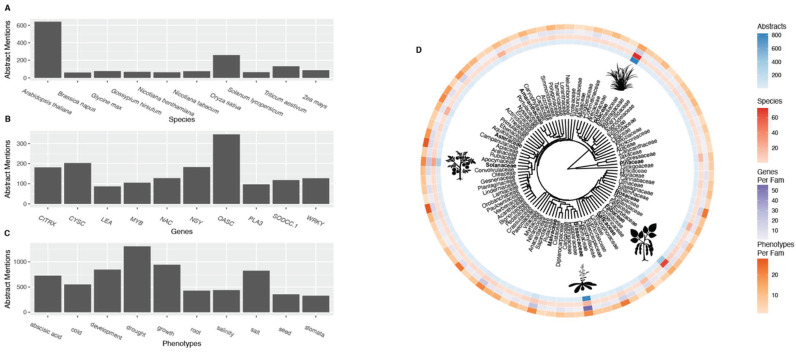
Most highly referenced (**A**) species, (**B**) genes, and (**C**) phenotype words within the consensus abstracts. (**D**) Phylogeny of families represented in the consensus abstracts shows that families with model or crop plants (e.g., Brassicaceae, Fabaceae, and Poaceae) were the most highly represented, both in terms of species included in studies and number of abstracts in which they were mentioned.

**Figure 2 ijms-23-12297-f002:**
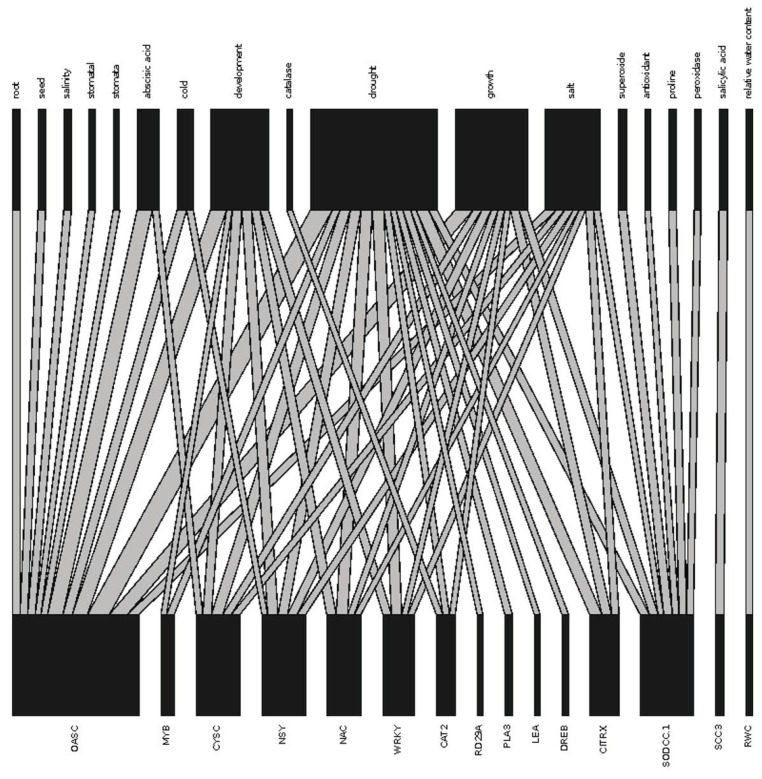
Bipartite plot showing the relationship between most referenced phenotypes (top row) and the most referenced genes with which they are associated (bottom row). The width of the block indicates the strength of the association for a given phenotype or gene and wider connections indicate more abstracts demonstrating that connection. The most highly mentioned phenotype word in the consensus abstracts was “drought”, which was highly associated with all top ten most highly mentioned genes, as well as *CAT2*, *RD29A*, and *DREB* genes.

**Figure 3 ijms-23-12297-f003:**
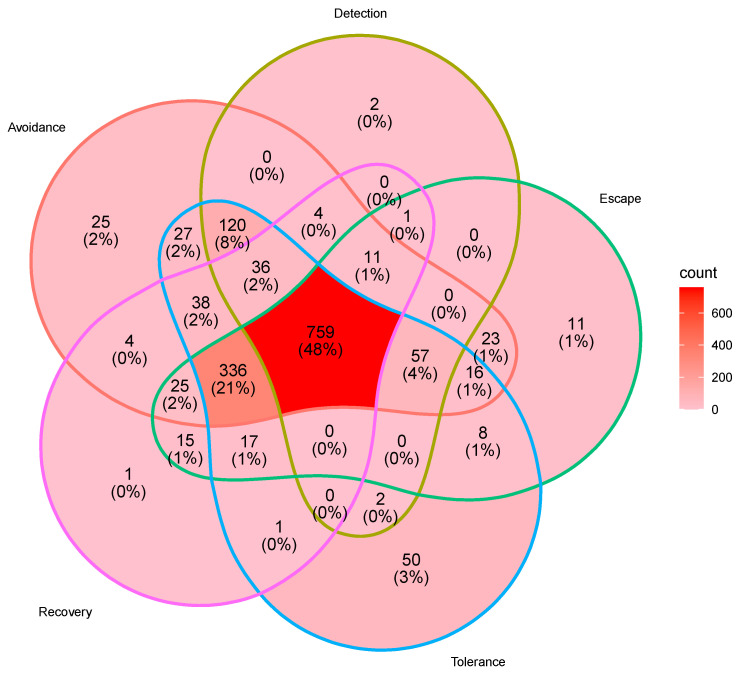
Venn diagram of the five major drought response strategies (avoidance, detection, escape, tolerance, and recovery) showing the overlap of genes associated with each category.

**Figure 4 ijms-23-12297-f004:**
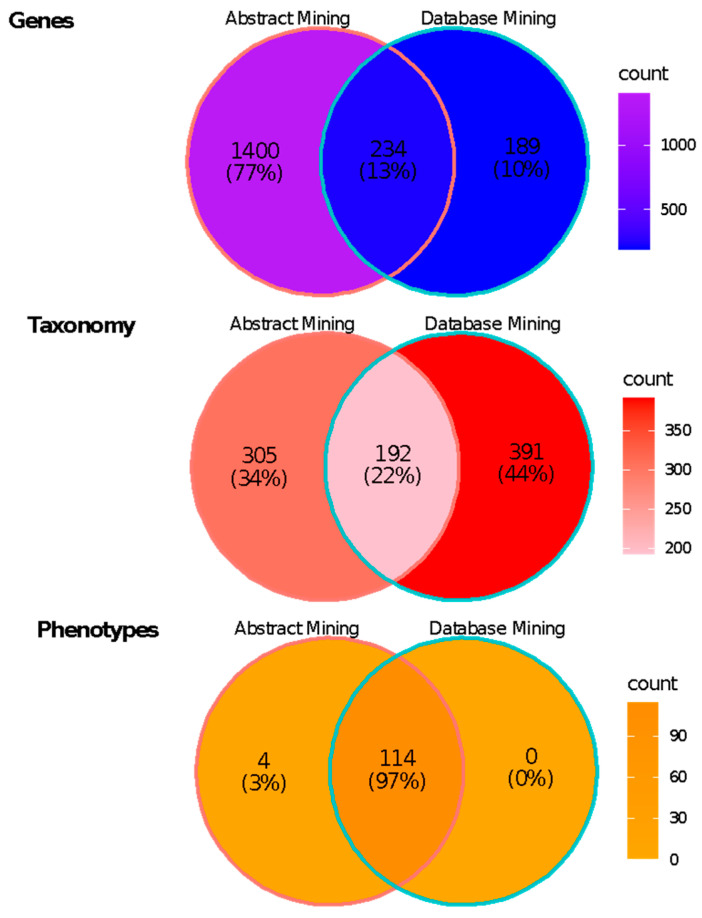
Overlap of genes, taxa, and phenotypes found using abstract mining with ‘G2PMineR’ and database mining with ‘rentrez’. Little overlap was found for gene and taxonomy data mined abstracts and databases (13% and 22%, respectively). Almost all phenotypes (97%) identified in the consensus abstracts were able to be identified in abstracts that were associated with gene and taxonomy data, as well.

**Figure 5 ijms-23-12297-f005:**
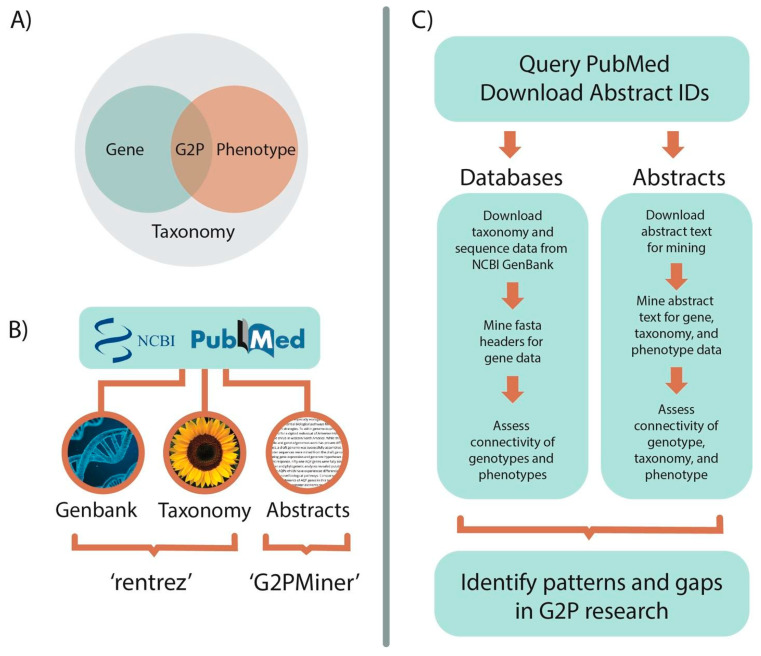
Workflow of abstract and database mining analyses. (**A**) The goal of these pipelines was to assess the connectivity of gene data to phenotype data all within a taxonomic framework, focusing on Viridiplantae. (**B**) NCBI data bases were chosen as the hub of this research as all data within these databases can be associated with an NCBI PubMed abstract ID. (**C**) Once abstracts of interest are identified, the PubMed IDs and abstract text can be downloaded for abstract and database mining analyses.

## Data Availability

The ‘G2PMineR’ R package is available at https://github.com/BuerkiLabTeam/G2PMineR. Supplemental R scripts for post-G2PMineR analyses are available via GitHub https://github.com/aemelton/G2PMineR_Supplemental_Scripts. Raw and quality-controlled occurrence data are available at DOI:10.5281/zenodo.7153278.

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
