# Peer review of "Meta-Analysis Reveals Challenges and Gaps for Genome-to-Phenome Research Underpinning Plant Drought Response"

_ijms, 2022, doi:10.3390/ijms232012297_

Round 1

Reviewer 1 Report

the comments have attached below

Author Response

Reviewer comment 1:

The “reviewed in…” refers to review style publication. This is not an uncommon way to let reviewers know which citations come from review articles instead of research articles. For example, https://doi.org/10.1111/1440-1703.12311 and 10.7717/peerj.11915 both use this type of notation.

Reviewer comment 2:

The parentheses around citations have been replaced with brackets.

Reviewer comment 3:

Figure 1 legend has been expanded to include an individual explanation for each part of the plot.

Reviewer minor concerns:

  • The abstract mining methods used for most downstream analyses are separate from databases and are thus unaffected by whether or not authors submitted data to a given database. Regarding analyses using database mining, we aimed to identify how well connect these publications are to the databases and do not require information other than a simple data available / unavailable for the analysis. Thus, in this instance our conclusion is dependent on whether or no data were deposited and not what has been or not been deposited.

  • The G2PMineR package has tools to curate gene lists and replace with accepted names from the SwissProt database. This has been clarified in the methods section.

Reviewer 2 Report

The study “Meta-analysis reveals challenges and gaps for genome-to-phenome research underpinning plant drought response” by Melton et al, is a great effort to compile research on stress on more than 497 plant species in various taxon.  This work is merited for publication in GENES – MDPI after some major modification. Therefore, the manuscript will be served as a beneficial source for the related plant field.

So, I have some points that may help to improve the work as follows:

-Abstract is good but needs more explanation about the main aim of the work

- The introduction should be extended to discuss the hypothesis and research questions in detail. Additionally, the introduction should cover the recent literature related to this subject.

Author Response

The study “Meta-analysis reveals challenges and gaps for genome-to-phenome research underpinning plant drought response” by Melton et al, is a great effort to compile research on stress on more than 497 plant species in various taxon.  This work is merited for publication in GENES – MDPI after some major modification. Therefore, the manuscript will be served as a beneficial source for the related plant field.

So, I have some points that may help to improve the work as follows:

-Abstract is good but needs more explanation about the main aim of the work

 The abstract has been modified to more explicitly state the aim of the research (lines 13 – 15).

- The introduction should be extended to discuss the hypothesis and research questions in detail. Additionally, the introduction should cover the recent literature related to this subject.

- More explicit hypotheses have been added to lines 84 – 98.

- We are unsure as to which subject in particular this comment refers to. We have review of the effects of anthropogenic climate change and plant response in lines, 28 – 53 (first two paragraphs of Introduction), Lines 54 – 70 review genomics and genomic resources for Genome-to-Phenome research, Lines 71 – 83 review public databasing advances challenges, and a supplemental review of drought response strategies is available in the Supplemental Appendix.

Reviewer 3 Report

General comments:

This is a very timely and interesting paper given that the topic of global climate change has got increasing attention from diverse fields. I don’t confess to be an expert in the fields of plant biology and bioinformatics, but the approach is easy to understand and looks logical. The discussion is very good. Each time I thought of something that might be missing, it showed up a few sentences later. Overall, the systematic approach, and statistical analysis is an approach that is appreciated, and results seem interesting. I have no objections to this part of the study.

I suggest that this paper be published after the authors solve the following minor suggestions.

Some minor observations:

1) To my knowledge, ‘meta-analysis’ is a proper noun and the meta-analysis has its own methodology (Gurevitch, J., Koricheva, J., Nakagawa, S., Stewart, G., 2018. Meta-analysis and the science of research synthesis. Nature 555(7695), 175-182). Please consider whether it is proper to call your study as a ‘meta-analysis’.

2) I am sorry but have to say that the coloration of Figure 2 seems a little ugly. Please select other colors rather than black and grey.

3) Conclusions should be revised. The present form seems more like an introduction but not conclusions. Please directly present your conclusions (maybe, and future directions).

4) R code. The analytical process is complex for me (and maybe some other readers). I notice that the R package ‘G2PMineR’ was developed by authors in 2021. This is wonderful. Can the authors upload a complete R code file as the supplementary material? This code file should be organized in the logical order of the manuscript and should be detailed annotated. Although the authors have uploaded their code to github, but the files are disordered. A detailed, annotated, logical code file can powerfully increase the influence of your study.

Author Response

This is a very timely and interesting paper given that the topic of global climate change has got increasing attention from diverse fields. I don’t confess to be an expert in the fields of plant biology and bioinformatics, but the approach is easy to understand and looks logical. The discussion is very good. Each time I thought of something that might be missing, it showed up a few sentences later. Overall, the systematic approach, and statistical analysis is an approach that is appreciated, and results seem interesting. I have no objections to this part of the study.

I suggest that this paper be published after the authors solve the following minor suggestions.

Some minor observations:

1) To my knowledge, ‘meta-analysis’ is a proper noun and the meta-analysis has its own methodology (Gurevitch, J., Koricheva, J., Nakagawa, S., Stewart, G., 2018. Meta-analysis and the science of research synthesis. Nature 555(7695), 175-182). Please consider whether it is proper to call your study as a ‘meta-analysis’.

We think that the use of “meta-analysis” is justified as we are largely analyzing the primary literature. There are additional analyses that do not directly analyze results reported in the literature, but all depend on the primary meta-analysis.

2) I am sorry but have to say that the coloration of Figure 2 seems a little ugly. Please select other colors rather than black and grey.

These are standard color schemes for bipartite plots. Since this is a personal preference not a critique of the science, we have chosen to keep the standard color scheme.

3) Conclusions should be revised. The present form seems more like an introduction but not conclusions. Please directly present your conclusions (maybe, and future directions).

The “Conclusions” section (lines 374-384) has been expanded to further describe our over-arching conclusions and our recommendation for future research foci.

4) R code. The analytical process is complex for me (and maybe some other readers). I notice that the R package ‘G2PMineR’ was developed by authors in 2021. This is wonderful. Can the authors upload a complete R code file as the supplementary material? This code file should be organized in the logical order of the manuscript and should be detailed annotated. Although the authors have uploaded their code to github, but the files are disordered. A detailed, annotated, logical code file can powerfully increase the influence of your study.

All scripts have been updated with more notation to explain each step and output. A more thorough README has been added explaining the order to run scripts in, which outputs will be used in other scripts, and the general workflow that they are used. I prefer to keep the scripts separate as this allows for containing code for a given purpose within one script and makes for easier debugging or improvement. The scripts and updated README will remain available on GitHub and not as a Supplemental File to reduce redundancy and ensure that interested parties access the same files.

Round 2

Reviewer 1 Report

The authors did a great revision work